# Selected Determinants of Diet Health Quality among Female Athletes Practising Team Sports

**DOI:** 10.3390/nu16193294

**Published:** 2024-09-28

**Authors:** Maria Gacek, Agnieszka Wojtowicz, Marlena Banasik

**Affiliations:** 1Department of Sports Medicine and Human Nutrition, Institute of Biomedical Sciences, University of Physical Education, 31-571 Krakow, Poland; 2Department of Psychology, Institute of Social Sciences, University of Physical Education, 31-571 Krakow, Poland; 3Department of Psychology, SWPS University of Social Sciences and Humanities—Jozef Tischner Campus, 31-864 Krakow, Poland

**Keywords:** nutrition of athletes, team sports, health locus of control, self-efficacy, personality traits, age, training experience

## Abstract

This study’s aim was an analysis regarding selected determinants of diet health quality in a group of elite Polish female team sport players. Relationships were assessed between age, sport experience, personal resources and personality traits with regard to the Big Five model and the pro-Health (pHDI-10) and non-Healthy (nHDI-14) Diet Indices. This study was conducted among 181 women (median age—25 years; sport experience—7 years) with the use of the Beliefs and Eating Habits Questionnaire (KomPAN), Generalised Self-Efficacy Scale (GSES), Multidimensional Health Locus of Control Scale (MHLC-B) and NEO-PI-R personality inventory. Statistical analysis was carried out via the Wilcoxon signed-rank test, Kruskal–Wallis’s ANOVA, Spearman’s rank correlation coefficient and forward stepwise regression at a significance level of α = 0.05. Multivariate regression analysis indicated that the value of the pro-Health Diet Index (pHDI-10) was positively explained by professional experience and extraversion, while negatively by openness to experiences (12% of the pHDI-10 variance). In turn, a higher value of the non-Healthy Diet Index (nHDI-14) was associated with the discipline of basketball (2% of the nHDI-14 variance). In summary, the demonstrated diet health quality was low and the predictive significance of competitive experience as well as type of discipline and selected personality traits was exhibited for diet quality among female team sport players.

## 1. Introduction

A diet that is varied and balanced, including products with a high nutritional value while limiting products of low nutritional density, helps maintain and improve health potential, optimise exercise capacity, effectively regenerate after exercise and reduce the risk of injury among athletes [1,2,3,4]. Increased nutritional needs also apply to athletes practising team sports, including women, whose nutritional needs are influenced by physiological factors, incorporating those that are hormonal and modulating, e.g., the course of metabolic processes [5,6,7,8,9,10,11,12,13]. At the same time, researchers draw attention to the limited number of scientific works on the nutrition of women practising sports [14]. The health quality of a diet is a function of implementing quantitative and qualitative nutritional recommendations, taking individual and environmental conditions into account, including, among others, age, sex, health state, physiological conditions, level of undertaken physical activity, etc. Various indicators related to the health quality of a diet can be found in the literature on the subject, whether nutrient-based, food/food group-based or combined, including, i.e., the Healthy Eating Index (HEI), Diet Quality Index (DQI), Healthy Diet Indicator (HDI), Mediterranean Eating Index (MEI), Australian Diet Quality Index (Aussie-DQI) or Athlete Diet Index (ADI) [15,16,17], but also described in the Polish literature are the pro-Health Diet Index (pHDI-10) and non-Healthy Diet Index (nHDI-14) [18,19]. The pro-Health Diet Index is defined by how frequently products having a potentially positive influence on health are consumed (vegetables, fruits, legumes, wholemeal bread, other whole-grain cereal products, milk, fermented dairy products, fromage frais, white meat and fish), while the non-Healthy Diet Index regards consuming products that can have a potentially adverse influence on health (white bread, other refined cereal products, fried foods, butter, lard, yellow and processed cheeses, meat products, red meat, canned meats, fast food, sweets, and sweetened, energy and alcoholic beverages) [18]. The health quality of athletes’ diets is of particular importance due to increased nutritional needs in conditions of intense physical effort, associated with intensified metabolic processes, which increase the demand for energy and for building and regulating components, as well as for fluids [1,2,3,5,7,9,10,11].

Nutritional behaviours are established by a broad range of individual as well as environmental factors [20,21]. In studies among athletes, significantly correlated variables—age and professional training experience—are indicated for the quality of nutritional choices, with an indication of more favourable ones for athletes with more sport experience [22]. A significant area concerning determinants related to nutritional choices is that of psychological factors, which include personal resources and personality traits. Among the personal resources important for health culture, a significant place is occupied by the locus of health control and sense of generalised self-efficacy, connected with the beliefs and expectations of an individual [23]. The health locus of control can be defined as the fairly steady belief of a person regarding factors that determine health, which are situated on a continuum from internal to external control. In athletes, achieving health goals determines optimal exercise capacity and is, therefore, one of the crucial factors of sport success. In previous studies, it was suggested that athletes with an internal locus of control declare a more rational way of eating than others who exhibit an external locus of control, although some results were not fully unambiguous; thus, the authors suggested further research in this area [22,24], which was one of the premises for undertaking the present study.

Another personal resource important for health culture is a sense of generalised self-efficacy. The concept of self-efficacy, created within the framework of A. Bandura’s social learning theory, conveys belief in the aptitude to reach anticipated goals, those related to complementary health and sports [23,25]. In previous studies, relationships were demonstrated between the sense of self-efficacy and the nutritional habits of competitive athletes, encompassing players of various team sports [22], as well as basketball and handball [26,27,28].

Personality traits create an internal regulatory system allowing adaptation to given situations as well as environments, and also internal integration concerning thoughts and behaviours [29]. One standard regarding personality is the five-factor (Big Five) model created by Costa and McCrae, which is one of the chief paradigms in trait psychology [30]. The five-factor model of personality comprises five main personality dimensions, them being neuroticism, extraversion, openness to experience, agreeableness and conscientiousness [31,32]. The indicated dimensions allow for a multi-aspect description and interpretation of personality with regard to five important areas, including emotional balance, general activity and contact, relations with other people, approach to duties, and overall attitude towards the world and new experiences [29,31]. Personality also has an influence on choices connected with nutrition quality. Such a situation was shown, i.e., in a study on Polish and Spanish students of physical education [33], as well as on males practising team sports [34,35,36]. Among P.E. students, an increase in the quality of a health-promoting diet was described in tandem with a rise in extraversion and conscientiousness levels [33]. The above-mentioned studies on the diet-related determinants of personality in athletes practising team sports showed, inter alia, that low neuroticism was conducive to the higher quality of a healthy diet and more beneficial nutritional behaviours with regard to exercise [34,36]. In some aspects of diet, the predictive significance of extraversion and conscientiousness for beneficial dietary choices was also described [34,35,36]. It should be added, however, that not all results were unambiguous, which suggested the need for further research also among female athletes, taking a wider range of variables into account, including age, sport experience and type of performed discipline [34,35,36]. This was another premise for undertaking the current study.

Assuming the key significance of the quality of diet for health and exercise competence, the multiplicity and complexity of dietary choices, the absence of exploitation concerning the issue of factors determining the diet of women professionally training for sports, suggestions appearing in the literature on the need for a multi-aspect assessment of athletes’ diets and the vagueness regarding the results of earlier studies, research was undertaken on a broader spectrum considering determinants of the diet among female athletes representing a high sport level. The study aim was to conduct an analysis of selected factors determining diet health quality among a group of elite Polish sportswomen training for team sports. The relationships between age, competitive experience in the sport, personal resources and personality traits included in the Big Five model and indices of diet health quality (pHDI-10 and nHDI-14) were evaluated. The following research questions were posed: (1) What are the health quality indices of the female athletes’ diet? (2) What are the personal resources (sense of health control and generalised self-efficacy) and personality traits (neuroticism, extraversion, openness to experience, agreeableness and conscientiousness) of female athletes? (3) What are the relationships between age, sport experience, personal resources and personality traits, as well as the quality of the female athletes’ diet? The research hypothesis was adopted suggesting that the analysed variables (demographic, sport and psychological) show a relationship with the health quality of the female athletes’ diet, with higher quality of the pro-health diet being supported by higher age and longer competitive experience, as well as more intense internal health control, higher sense of self-efficacy and extraversion level (related to cheerfulness), conscientiousness (related to dutifulness) and a lower neuroticism level (related to less immoderation).

## 2. Material and Methods

### 2.1. Participants

The research was carried out between 2021 and 2024 in a group of 181 women training in team sports, including football (*n* = 36), handball (*n* = 68), volleyball (*n* = 45) and basketball (*n* = 32). The statistically estimated group size in the G*Power programme for 4 sport disciplines, at a level of statistical significance totalling 0.05 and power equalling 0.95, should be from 112 (for strong effects) to 280 individuals (for medium-sized effects). The basic criteria used to select women for the study group was for them to be training a team sport discipline at a competitive level, i.e., the highest possible league level in Poland, and having a minimum of 3 years professional experience. The exclusion criteria from the group were male gender, low sport level, less than 3 years of professional experience and a break in training due to injury. The athletes under study, with regard to the present classification of activity level and sport capabilities [37], could be allocated to Tier 3 (Highly Trained or National Level). This study was performed according to the principles included in the 1964 Declaration of Helsinki. Informed consent was obtained from the respondents. The research protocol was approved by the Bioethics Committee at the District Medical Chamber in Kraków (No. 105/KBL/OIL/2021, dated 16 April 2021).

The socio-demographic and sport data of the group collected in the survey allowed the indication that the studied sportswomen were aged 18 to 33 (Me = 25 years), with their sport experience being within the range of 3 and 15 years (Me = 7 years). They undertook an average of 7 training sessions per week (Table 1). The level of education and place of residence of the women representing individual disciplines did not differ significantly (Table 2 and Table 3).

### 2.2. Instruments

#### 2.2.1. Assessment of Diet Health Quality

The Beliefs and Eating Habits Questionnaire (KomPAN), designed by the Committee of Human Nutrition Science at the Polish Academy of Sciences [18], was implemented to assess the diet. It shows a sufficiently high repeatability of results [38]. The frequency of product consumption was evaluated on a scale from 0 to 6, with the following ranks being adopted: (6) several times a day, (5) once a day, (4) several times a week, (3) once a week, (2) sporadically, i.e., 1–3 times a month, (1) never. Then, these ranks were converted into real numbers, meaning the daily frequency of consumption (times/day), according to the scheme: (2) several times a day, (1) once a day, (0.5) several times a week, (0.14) once a week, (0.06) several times a month, (0) never [18]. The scale does not take the volume/grammage of the consumed foods into account. The pro-Health (pHDI-10) and non-Healthy (nHDI-14) Diet Indices were calculated as the sum of the daily frequency of consuming appropriate products. The pHDI-10 is established on the basis of how frequently 10 selected product groups are consumed (vegetables, fruits, legumes, wholemeal bread, other whole-grain cereal products, milk, fermented dairy products, fromage frais, white meat and fish). The nHDI-14 is determined with regard to the consumption frequency of 14 product groups (white bread, other refined cereal products, fried foods, butter, lard, yellow and processed cheeses, meat products, red meat, canned meats, fast food, sweets, and sweetened, energy and alcoholic beverages) [18]. The indices are interpreted in such a way that the higher the index value, the greater the nutritional features’ intensity, whether beneficial or adverse for health. The pHDI-10 values, given as times/day, were within the range of 0–20, and the nHDI-14 values were between 0 and 28. The pHDI-10 values within the range of 0–6.66 were assessed as low, between 6.67 and 13.33 was considered moderate and the range of 13.34–20.0 was assumed as high. The nHDI-14 values within the range of 0–9.33 were described as low, 9.34–18.66 as moderate and 18.67–28.0, high. To simplify interpretation, the values of both indices expressed as times/day were converted to a 0–100-point scale (with the interpretation: 0–33 low, 34–66 moderate and 67–100 high) according to the described methodology [18]. Dietary quality index profiles (DQIPs) were also calculated. Based on the calculations, with the procedure [18,19], 5 indicators were obtained, i.e., 3 pHDI-10 indicators of three level of adherence to a pro-healthy diet—ow, moderate and high; and 2 nHDI-14 indicators of the two level of adherence to a non-healthy diet—low and moderate. In order to identify a specific structure of the quality of the diet, 6 diet quality index profiles were developed. They were made up of combinations of the 5 indicators (3 pHDI-10 × 2 nHDI-14). According to the methodology [18,19], the possible profiles of diet quality indicators are as follows: DQIP-1: low pHDI-10 and low nHDI-14; DQIP-2: low pHDI-10 and moderate nHDI-14; DQIP-3: moderate pHDI-10 and low nHDI-14; DQIP-4: moderate pHDI-10 and moderate nHDI-14; DQIP-5: high pHDI-10 and low nHDI-14; DQIP-6: high pHDI-10 and moderate nHDI-14. This methodology was also described and used in another population group in Poland [19].

#### 2.2.2. Assessment of Generalised Self-Efficacy

Self-efficacy was evaluated via the Generalised Self-Efficacy Scale (GSES) by R. Schwarzer, M. Jerusalem and Z. Juczyński [23]. The obtained GSES scores were between 10 and 40 points (the higher the score, the greater the sense of generalised self-efficacy). The GSES indicated a high level of internal consistency (Cronbach’s α-coefficient was at the level of 0.85) [23].

#### 2.2.3. Assessment of Health Locus of Control

The health locus of control was established using the standardised Multidimensional Health Locus of Control Scale (MHLC-B), created by Wallston et al. [39], in the adapted Polish version by Z. Juczyński [23]. The MHLC-B scale consists of 18 diagnostic statements describing 3 dimensions related to the control health locus: internal (I), influence of others (P) and chance (C). The internal dimension (I) denotes control over health dependent on the given individual, the second one regards the strong influence of other people (P), including medical staff, and the third one refers to the influence of external factors, chance (C) [23]. It is assumed that the influence of others and chance make up the external health locus of control. The scale comprises 6 assessment degrees regarding specific items, ranging from “I absolutely disagree” (1) to “I absolutely agree” (6). The results of each 3 subscales are within the range of 6–36 points, and the higher the achieved score, the stronger the given dimension of the control health locus is. The MHLC-B scale demonstrated high psychometric properties. Version B’s reliability was 0.64 according to Cronbach’s alpha coefficient in the case of internal control, 0.63 with regard to chance and 0.59 in relation to influence of others [23].

#### 2.2.4. Assessment of Personality Traits from the Five-Factor Model

The NEO-PI-R (Neuroticism Extraversion Openness Personality Inventory—Revised) by Costa and McCrae [31] in its Polish adaptation [32] was implemented for assessment of personality traits. The NEO-PI-R Personality Inventory includes 240 statements, with a 5-point response scale (ranging from “I completely disagree” to “I completely agree”). The statements are an expression of 5 personality dimensions: neuroticism, extraversion, openness to experience, agreeableness and conscientiousness. The measurement reliability of the Polish-adapted version of the questionnaire was sufficient and reached: 0.86 for neuroticism, 0.85 for extraversion, 0.86 for openness to experience, 0.81 for agreeableness and 0.85 for conscientiousness [32].

### 2.3. Statistical Analyses

Statistical analyses were conducted via Statistica 13.1. Basic descriptive statistics of the quantitative variables under examination (minimum, maximum, mean, standard deviation, median, quartiles) were evaluated. The distributions of the analysed variables deviated from normality; therefore, the median was chosen as the measure of central tendency. Friedman’s non-parametric analysis of variance was used to detect differences in the level of types of control health locus and the level of personality traits, and the non-parametric Wilcoxon test was used to establish potential differences related to the level of dietary health quality indices. Kruskal–Wallis’s non-parametric ANOVA with z tests was used to determine differences between individual disciplines in the level of MHLC, GSES, indices of dietary health quality and personality traits. Spearman’s non-parametric correlation coefficient was implemented to establish relationships between dietary quality indices and MHLC, GSES, personality traits, age and experience of the athletes. Forward stepwise regression was used to describe the models, explaining the level of dietary quality indices. The adopted level of statistical significance was α = 0.05.

## 3. Results

### 3.1. Diet Health Quality of Female Athletes Training in Team Sports

The most frequently consumed foods by the respondents (at least once a day) were fruits (47.5%), vegetables (36.4%), meat products (57.5%), white bread (34.8%) and sweets (29.3%) (Table 4).

In Table 5, the frequency of consuming selected product groups per day and the values of the diet quality indices are presented. Among the products that are recommended in the diet, the athletes most frequently ate fruits and vegetables, and least frequently, fermented dairy products. With regard to the products having potentially harmful effects on health, the respondents most frequently consumed processed meats, white bread, butter, sweets and confectionery, and least frequently, canned meats, as well as fried and fast foods. The pHDI-10 and nHDI-14 values were 17.70 and 14.64 points on a unified point scale. The analyses allowed the indication that the level of the pro-Health Diet Index (pHDI-10) was significantly greater than the non-Healthy Diet Index (nHDI-14) (T = 4935.00, Z = 4.68, *p* < 0.001) (Table 5).

The profiles of the diet quality index of the women representing individual disciplines did not differ significantly (Table 6). Only two female athletes (volleyball players) had a better diet profile than the rest of the participants.

### 3.2. Position of Health Control (MHLC), Generalised Self-Efficacy (GSES) and Personality Traits in the Five-Factor Model among Female Athletes Practising Team Sports

Among the examined female athletes, the internal health locus of control was dominant, and the lowest level was described for the influence of chance (ANOVA chi-square = 280.49, df = 2, *p* < 0.001). The level regarding sense of generalised self-efficacy among the female athletes reached 32 points. Differences were found in the level of particular personality traits for the female athletes in the five-factor model (ANOVA chi-square = 344.83, df = 4, *p* < 0.001). The highest level was achieved by conscientiousness (Me = 133.0), then agreeableness (127.0). In third place, extraversion (Me = 117.0) and openness (Me = 116.0) were found, and the lowest level was obtained by neuroticism (Me = 75.0) (Table 7).

No statistically significant differences were observed with regard to the level of MHLC P between disciplines (H_3,n=181_ = 4.66, *p* = 0.199); however, the discipline groups differed in terms of MHLC I (H_3,n=181_ = 20.99, *p* < 0.001) and MHLC C (H_3,n=181_ = 9.40, *p* = 0.024). It was found that the volleyball players had a lower-level internal health locus of control than the basketball, handball and football players, although, in this instance, the value was *p* = 0.071 (Table 8), while the handball players demonstrated a lower level of health control given to chance than the basketball players (Table 9).

Differences were also observed between the disciplines in the level of generalised self-efficacy (H_3,n=181_ = 16.09, *p* = 0.001). Handball players exhibited a higher level of generalised self-efficacy than football or basketball players (Table 10).

There was a lack of statistically significant differences between the disciplines in the level of individual personality traits: neuroticism (H_3,n=181_ = 1.37, *p* = 0.712), extraversion (H_3,n=181_ = 0.68, *p* = 0.877), openness (H_3,n=181_ = 0.78, *p* = 0.854), agreeableness (H_3,n=181_ = 6.81, *p* = 0.078) or conscientiousness (H_3,n=181_ = 4.52, *p* = 0.210).

### 3.3. Diet Health Quality of Female Athletes Training in Team Sports Depending on Performed Discipline and Locus of Health Control (MHLC), Generalised Self-Efficacy (GSES), Personality Traits of the Big Five, Age and Training Experience

There were no statistically significant differences in the pHDI-10 level between the disciplines (H_3,n=181_ = 2.58, *p* = 0.460), but these groups differed in terms of the nHDI-14 (H_3,n=181_ = 7.99, *p* = 0.046). It was found that female football players exhibited a significantly lower level in the non-Healthy Diet Index than their peers playing basketball (Table 11).

Analyses of correlations between indices regarding a healthy diet quality (pHDI-10 and nHDI-14), as well as health locus of control, generalised self-efficacy, age and professional training experience, did not show any statistically significant relationships, although a tendency for the pro-Health Diet Index to increase, along with the rise of the internal control health locus, age and professional experience, could be observed (Table 12). In the case of relationships with the personality traits of the female athletes, it was shown that along with a rise in openness, the pro-Health Diet Index (pHDI-10) decreased. A statistical tendency for the pro-Health Diet Index to increase with the rise of conscientiousness was also demonstrated (Table 12).

Forward stepwise regression analysis showed that in the model explaining the pro-Health Diet Index, three statistically significant predictors remained, which explained 12% of the variance for the pHDI-10 (r = 0.34, r^2^ = 0.12, F(3,177) = 7.98, *p* < 0.001). A positive relationship between the pHDI-10 and professional sport experience as well as level of extraversion, and a negative relationship between the pHDI-10 and the level of openness were found (Table 13).

Forward stepwise regression analysis showed that in the model explaining the non-Healthy Diet Index, there remained one statistically significant variable that explained 2% of the nHDI-14 variance (r = 0.15, r^2^ = 0.02, F(1,179) = 3.97, *p* = 0.048). The female basketball players showed a higher level of the nHDI-14 (Table 14).

## 4. Discussion

In the discussed study, a low level was demonstrated regarding the health quality of the diet. The dominance of the internal health locus of control, a high level of generalised self-efficacy and conscientiousness and, further, a low level of neuroticism were noted. Significant relationships were further observed between selected demographic, sport and psychological variables and indices related to the health quality of the diet among women professionally practising team sports at a high level. The pro-Health Diet Index (pHDI-10) was shown to increase with experience and extraversion and to decrease with openness to experience, while the non-Healthy Diet Index (nHDI-14) was particularly associated with basketball. The obtained results allowed for partial positive verification of the adopted research hypothesis.

### 4.1. Diet Quality of the Female Athletes

Among the studied Polish elite female team sport players, a low level of healthy diet quality was demonstrated, with low values recorded for both the pro-Health (pHDI-10) (17.70 points) and non-Healthy (nHDI-14) (14.64 points) Diet Indices, which should be interpreted as having low, beneficial and detrimental influences of the implemented diet on the training women’s health. The low level of the pro-Health Diet Index resulted from the low consumption of products recommended, inter alia, those rich in antioxidants (fruits and vegetables), dietary fibres (low-grain cereals, legumes), probiotics (fermented dairy products) and omega-3 PUFA (sea fish). The low consumption of these products reduced the health quality of the diet. The indicated nutrients constitute functional foods, important for improving health and preventing chronic diseases [40,41,42,43]. A diet that is rich in antioxidant vitamins and polyphenols as well as alkalizing and probiotic products is also of extreme significance for athletes, as it helps restore physiological homeostasis disturbed in conditions of intense physical effort, including reducing oxidative stress, restoring acid–base balance, and limiting the immediate and distant effects of intestinal dysbiosis [41,44,45,46,47,48]. The exhibited low level of the pro-Health Diet Index in elite Polish female team sport players is consistent with the results obtained in this regard for elite Polish athletes (men) professionally training in team sports, who also had low values of pro-Health and non-Healthy Diet Indices (19.16 and 15.69 points) [34]. A low diet health quality was also described among young Brazilian athletes, including female volleyball players [49]. The low diet health quality of the examined female team sport players was also in accordance with the trends described in other groups of team sport athletes, who were characterised as making poor nutritional choices and incorporating a poorly balanced diet [5,26,27,28,36,50,51,52,53,54,55,56,57,58,59,60]. Different results indicating a high pro-health quality of diet were shown in Australian individual and team sport players [61].

### 4.2. Psychological Traits of the Female Athletes

It should be noted that the basic analysis of the female athletes’ personality profiles was conducted to constitute only a backdrop for the evaluation of potential relationships between psychological traits and diet health quality, and therefore, the scope was limited. The discussed research allowed the observation that in the elite group of Polish team sport players, in terms of the analysed psychological traits, a dominant internal locus of health control was characteristic, as well as a high sense of generalised self-efficacy (seventh sten according to sten norms) [23], and also a low level of neuroticism (third sten), an average level of extraversion (sixth sten) and openness to experience (fifth sten), and a high level of agreeableness and conscientiousness (seventh sten) (according to sten norms for women aged 17–29 for the NEO-PI-R inventory) [32]. It was also demonstrated that the type of discipline differentiated some personality traits, with an indication towards female handball players, who in the group were distinguished by low external health control situated in chance and a higher efficacy level. It can thus be stated that the studied group of team sport players was dominated by women who were convinced of their own influence on their health, expressed a positive belief in the prospect of obtaining their given goals and were emotionally stable, with a high level in the conscientiousness dimension, which included, among others, striving for achievement, self-discipline and a sense of duty. The occurrence of the dominant internal locus of control and a high generalised sense of self-efficacy in the studied players refers to the theory that the internal locus of control for health is influenced by various factors, including a high sense of self-efficacy and health’s high position in their value system [25]. Another group of team sport athletes also exhibited the highest level of internal locus of control for health (Me = 27), the lowest in the case of chance (Me = 18) and a high self-efficacy level (Me = 31) [22]. In terms of generalised self-efficacy level, other athletes practising handball and basketball also presented a high level of this trait (Me = 31–32) [26,27]. In previous studies, it was shown that male athletes who practise team sports also had a partially similar personality profile, as they were categorised by high levels of conscientiousness, agreeableness, extraversion and openness, and low levels of neuroticism [34]. Low levels of neuroticism from the five-factor model of personality traits among athletes were also confirmed in other papers [62,63,64]. In a systematic review of the literature, it was specified that all personality traits (except for neuroticism) were conducive to achieving sport success [65].

### 4.3. Demographic, Sport-Related and Psychological Conditions of the Female Athletes’ Diet Quality

In the current research, significant differentiation was observed for diet health quality depending on the practised sport discipline, with an indication of lower diet quality (higher non-Healthy Diet Index—nHDI-14) in female basketball players compared to female football representatives. A statistical trend towards better diet health quality (higher pHD-10 level) was also found with regard to age, professional experience and internal health locus of control in female players. No significant correlations were noted between generalised self-efficacy and diet health quality. However, it was also demonstrated that diet quality decreased along with an increase in openness to experience. Multivariate forward stepwise regression analysis made it possible to validate the predictive significance of several analysed variables in explaining the variance regarding diet health quality indices. The value of the pro-Health Diet Index (pHDI-10) was positively explained by competitive experience and extraversion, and negatively by openness to experience (12% of the pHDI-10 variance). In turn, a higher value of the non-Healthy Diet Index (nHDI-14) was associated with the discipline of basketball (2% of the nHDI-14 variance).

The positive correlation between competitive experience and higher diet health quality among the studied female athletes referred to the results obtained in earlier trials, in which it was shown that age and training experience (positively correlated) significantly differentiated selected aspects concerning the diet of athletes practising team sports. This further indicated a positive correlation between age and the more frequent consumption of products recommended in the diet, comprising, among others, vegetable juices and mineral water, medium- and low-glycaemic products (vegetables, coarse-grained groats, legumes, low-fat dairy products) and products rich in unsaturated fatty acids (fish, almonds, olive oil), with a simultaneous reduction in the frequency of consuming high-glycaemic, highly refined cereal products, as well as atherogenic animal fats and trans isomers [22].

In the case of relationships between diet health quality and personality traits, the confirmed predictive significance of extraversion and negative significance of openness to experience for the pro-Health Diet Index were also justified in the context of previous studies. For example, in research among elite athletes (men) training in team sports, significant relationships were indicated between the personality traits found in the Big Five model (extraversion and neuroticism) and indices of diet health quality. In the cited studies conducted among training males, similarly to training women, high extraversion was associated with a higher pro-Health Diet Index (pHDI-10), and with a greater incidence of consuming products with high nutritional value, i.e., vegetables and dairy products [34]. Partially similar relationships were also described among academic youth undertaking an increased level of physical activity (P.E. students from Poland and Spain). In these individuals, both the indices of the pHDI-10 and nHDI-14 rose with the intensification of extraversion (while in the studied female athletes, only a change in the pHDI-10 was noted). In P.E. students, it was also demonstrated that high extraversion was associated with consuming fruit and vegetables, and furthermore, with sweets and confectionery as well [33]. The positive significance of extraversion for fruit consumption was also discussed among academic youth from New Zealand [66]. In terms of another personality trait predictive of the pro-Health Diet Index—openness to experience—no significant relationships between these variables were exhibited in other studies carried out among athletes practising team sports [34]. In another study, however, it was shown that with the intensification of openness to experience, the index of correct nutritional behaviours referring to the recommendations of the Swiss nutrition pyramid for athletes significantly decreased [35], which corresponded to the discussed results among female team sport athletes. In another cohort of athletes—males training in team sports—similarly to the studied women, the model explaining the value obtained for the pro-Health Diet Index also included extraversion (a positive relationship) [34].

The obtained results indicating a low diet health quality (connected with a higher non-Healthy Diet Index—nHDI-14) in female basketball players corresponded with the lack of employing a balanced diet, indicated in previous studies among athletes training in basketball, also at a high sport level. A quantitative assessment regarding the dietary choices of elite basketball players showed a low intake of dietary fibres and of some vitamins (D, E, B9, B1, C) and minerals (potassium, calcium, magnesium) [26]. In other studies, a limited scale was indicated regarding the implementation of qualitative nutritional recommendations among basketball players [28]. The described trends clearly showed reduced nutritional quality and diet health quality at the same time, corresponding with the results obtained for the examined female players.

In contrast to the present study, in other groups of athletes, significant correlations were demonstrated between personal resources, including sense of self-efficacy, locus of control and diet quality. The lack of significant correlations among the examined women may have been due to the generally low level of the pro-Health Diet Index and generally high sense of self-efficacy among the female athletes. In other studies on the subject, positive predictive significance was confirmed concerning the sense of generalised self-efficacy for the quality of athletes’ diet, including elite Polish basketball [26,28] and handball players [27]. Research on correlations between the health locus of control and nutritional choices of athletes are less numerous; however, the statistical tendency found in the current research towards a higher quality of a healthy diet among individuals with a stronger internal sense of health control corresponded to the results of previous research conducted among athletes practising team sports. In this research, it was observed that a high internal health locus of control was linked with rational nutritional choices, including more frequent consumption of recommended sources of full-value proteins and vegetable fats, as well as a preference for mineral water and avoidance of sweetened carbonated and flat beverages [22]. More rational nutritional choices among athletes with an internal, instead of external, locus of control were also noted in other studies [24].

In general, it can be stated that research on the determinants of diet quality may generate ambiguous results. However, the obtained results may not only be of cognitive value but can also be of applicative significance in shaping proper nutritional choices as an important element of sport success, taking demographic, sport and psychological conditions into account. Further advanced interdisciplinary studies on explaining the mechanisms of the described relationships would be justified, as also pointed out by other authors [67]. A low diet health quality was demonstrated in female athletes training in team sports, which considered the individualisation of interactions promoting proper nutritional choices, as also indicated by other researchers [50,60,68,69,70].

### 4.4. Limitations

The limitations of the presented study mainly regard the consideration of one particular area of nutrition (diet quality) and the self-descriptive nature of the implemented research. It should further be emphasised that the obtained results concern only women (the results for men from respective sport groups are the subject of other previous works included in the list of references). The highlighted limitations, as well as others, may be grounds for specifying further directions of research. Additional studies should be focused on comprehensively assessing personality determinants from various perspectives and with regard to the nutrition of females actively taking part in sports. Further research in this area may concern, among others, psychological determinants of nutritional behaviours and quantitative aspects of diet among female athletes.

## 5. Conclusions

Poor diet health quality was demonstrated in elite Polish female athletes training in team sports, with the described level of the pro-Health and non-Healthy Diet Indices being low, suggesting a limited, positive and detrimental influence of the applied diet on health.

Elite female team sport players, among the analysed psychological traits, were characterised by the dominance of their internal health locus of control, high generalised sense of self-efficacy, and also a low level of neuroticism, an average level of extraversion and openness to experience, and a high level of agreeableness and conscientiousness.

Demographic, sport and psychological traits were shown to be predictive of diet quality in female team sport players, with the complex model explaining the diet quality indices confirming that the level of the pro-Health Diet Index was positively explained by competitive experience and extraversion, and negatively by openness to experience. In contrast, a higher value of the non-Healthy Diet Index was characteristic of female basketball players. The predictive significance of sport and psychological characteristics for the diet health quality among women practising team sports was validated.

The results achieved in this study allowed an indication of the validity of monitoring diet and undertaking educational activities to increase diet health quality among professional female athletes, taking demographic, sport and psychological characteristics into account.

## Figures and Tables

**Table 1 nutrients-16-03294-t001:** Demographic and sport characteristics of the female athletes (descriptive statistics, *N* = 181).

Variables	Mean	SD	Median	Min.	Max.	Q25	Q75
Age (years)	25.27	3.81	25.00	18.00	33.00	22.00	29.00
Experience (years)	7.27	3.81	7.00	0.00	15.00	4.00	11.00
No. of training sessions/week	6.77	1.15	7.00	4.00	8.00	7.00	7.00

SD—standard deviation; Q25—lower quartile; Q75—upper quartile.

**Table 2 nutrients-16-03294-t002:** Sport discipline and education of the female athletes (descriptive statistics and chi-square test, *N* = 181).

Discipline	Education				Total
	Secondary		Higher		
	*N*	%	*N*	%	*N*
Football	13	36.11	23	63.89	36
Basketball	10	31.25	22	68.75	32
Handball	15	22.06	53	77.94	68
Volleyball	19	42.22	26	57.78	45
Total	57	31.49	124	68.51	181

Chi-square = 5.56; df = 3; *p* = 0.135.

**Table 3 nutrients-16-03294-t003:** Discipline and place of the female athletes’ residence (descriptive statistics and chi-square test, *N* = 181).

Discipline	Place of Residence								Total
	Village		City < 20,000		City 20,000–100,000		City > 100,000		
	*N*	%	*N*	%	*N*	%	*N*	%	*N*
Football	9	25.00	12	33.33	11	30.56	4	11.11	36
Basketball	3	9.38	6	18.75	16	50.00	7	21.88	32
Handball	7	10.29	17	25.00	27	39.71	17	25.00	68
Volleyball	6	13.33	12	26.67	17	37.78	10	22.22	45
Total	25	13.81	47	25.97	71	39.23	38	20.99	181

Chi-square = 9.67; df = 9; *p* = 0.378.

**Table 4 nutrients-16-03294-t004:** Frequency of consuming products having potentially positive (pHDI-10) and negative (nHDI-14) influence on health in the studied female athletes (*N* = 181, percentage of respondents).

Indices/Products		Frequency of Consumption					
		1	2	3	4	5	6
		% Respondents					
Potentially beneficial effects on health (pHDI-10)	Fruit	0.0	0.0	17.1	35.4	39.8	7.7
	Vegetables	0.0	4.4	23.8	35.4	33.1	3.3
	Whole-wheat bread	13.3	34.8	32.6	17.7	1.7	0.0
	Coarse grains, oatmeal, whole-grain pasta	2.2	12.7	44.8	40.3	0.0	0.0
	Legume seeds	6.6	28.2	40.3	19.9	5.0	0.0
	Milk	6.6	26.0	27.1	23.8	11.0	5.5
	Fermented dairy products	14.9	40.9	33.7	9.4	1.1	0.0
	Fromage frais	1.7	18.8	45.3	24.3	8.8	1.1
	Poultry dishes	0.0	12.7	44.8	23.8	18.8	0.0
	Fish dishes	2.2	22.1	29.8	39.2	6.6	0.0
Potentially detrimental effects on health (nHDI-14)	White bread	6.6	9.4	19.3	29.8	17.7	17.1
	White rice, small groats	1.1	30.9	36.5	31.5	0.0	0.0
	Yellow, blue and processed cheeses	4.4	40.9	41.4	9.9	3.3	0.0
	Cold-cuts, sausages and hotdogs	0.0	0.0	11.0	31.5	35.4	22.1
	Red meat dishes	1.1	22.1	48.1	21.0	7.7	0.0
	Canned meats	18.8	45.9	28.2	6.1	1.1	0.0
	Fried dishes	16.0	53.6	11.0	19.3	0.0	0.0
	Butter	10.5	9.9	28.7	26.0	13.3	11.6
	Lard	94.5	5.5	0.0	0.0	0.0	0.0
	Fast food	31.5	49.7	18.8	0.0	0.0	0.0
	Sweets and confectionery	1.7	5.5	27.1	36.5	12.7	16.6
	Sweetened carbonated and flat beverages	18.8	23.8	35.9	12.7	8.8	0.0
	Energy drinks	14.9	24.3	34.8	22.1	3.9	0.0
	Alcoholic beverages	22.1	58.0	17.7	2.2	0.0	0.0

Legend: (1) never, (2) 1–3 times a month, (3) once a week, (4) several times a week, (5) once a day, (6) several times a day.

**Table 5 nutrients-16-03294-t005:** Daily frequency of consuming products having potentially positive and negative influence on health and values of pro-Heath (pHDI-10) and non-Healthy Diet (nHDI-14) Indices in the studied female athletes.

Products/Indices		M	SD	Me	Min	Max	Q25	Q75
Potentially beneficial effects on health (pHDI-10)	Fruit	0.75	0.48	0.50	0.14	2.00	0.50	1.00
	Vegetables	0.61	0.43	0.50	0.06	2.00	0.14	1.00
	Whole-wheat bread	0.17	0.20	0.14	0.00	1.00	0.06	0.14
	Coarse grains, oatmeal, whole-grain pasta	0.27	0.19	0.14	0.00	0.50	0.14	0.50
	Legume seeds	0.22	0.24	0.14	0.00	1.00	0.06	0.32
	Milk	0.39	0.49	0.14	0.00	2.00	0.05	0.50
	Fermented dairy products	0.13	0.16	0.06	0.00	1.00	0.06	0.14
	Fromage frais	0.31	0.33	0.14	0.00	2.00	0.14	0.50
	Poultry dishes	0.38	0.34	0.14	0.06	1.00	0.14	0.50
	Fish dishes	0.32	0.27	0.14	0.00	1.00	0.14	0.50
Potentially detrimental effects on health (nHDI-14)	White bread	0.70	0.67	0.50	0.00	2.00	0.14	1.00
	White rice, small groats	0.23	0.19	0.14	0.00	0.50	0.06	0.50
	Yellow, blue and processed cheeses	0.17	0.20	0.14	0.00	1.00	0.06	0.14
	Cold-cuts, sausages and hotdogs	0.97	0.62	1.00	0.14	2.00	0.50	1.00
	Red meat dishes	0.26	0.27	0.14	0.00	1.00	0.14	0.50
	Canned meats	0.11	0.15	0.06	0.00	1.00	0.06	0.14
	Fried dishes (flour and meat)	0.14	0.18	0.06	0.00	0.50	0.06	0.14
	Butter	0.54	0.61	0.50	0.00	2.00	0.14	0.75
	Lard	0.00	0.01	0.00	0.00	0.06	0.00	0.00
	Fast food	0.06	0.05	0.06	0.00	0.14	0.00	0.06
	Sweets and confectionery	0.68	0.65	0.50	0.00	2.00	0.14	1.00
	Sweetened carbonated and flat beverages	0.22	0.29	0.14	0.00	1.00	0.06	0.14
	Energy drinks	0.21	0.24	0.24	0.00	1.00	0.06	0.50
	Alcohol	0.07	0.08	0.08	0.00	0.50	0.06	0.06
Indices	pHDI-10 (times/day)	3.55	3.13	2.04	0.26	13.50	1.43	5.10
	nHDI-14 (times/day)	4.36	4.21	3.56	0.14	14.70	1.48	5.93
	pHDI-10 (points)	17.77	4.69	17.70	8.60	33.90	14.50	21.00
	nHDI-14 (points)	15.57	5.28	14.64	5.29	28.71	12.07	18.64

Legend: M—arithmetic mean; SD—standard deviation; Me—median; Q25—lower quartile; Q75—upper quartile.

**Table 6 nutrients-16-03294-t006:** Diet quality index profiles (DQIPs) with the proportion of study participants in each group.

Discipline	Diet Quality Index Profiles				Total
	DQIP-1 (Low pHDI-10 and Low nHDI-14)		DQIP-3 (Moderate pHDI-10 and Low nHDI-14)		
	*N*	%	*N*	%	*N*
Football	36	100.00	0	0.00	36
Basketball	32	100.00	0	0.00	32
Handball	68	100.00	0	0.00	68
Volleyball	43	95.56	2	4.44	45
Total	179	98.90	2	1.10	181

Chi-square = 5.63; df = 3; *p* = 0.131.

**Table 7 nutrients-16-03294-t007:** Psychological characteristics of the female athletes—position of health control (MHLC), generalised self-efficacy (GSES) and personality traits—in accordance with the Big Five model (descriptive statistics, *N* = 181).

Variables	Mean	SD	Median	Min.	Max.	Q25	Q75
MHLC I	27.09	2.83	27.00	20.00	32.00	25.00	29.00
MHLC P	20.93	2.58	21.00	16.00	26.00	19.00	23.00
MHLC C	17.03	3.21	17.00	11.00	26.00	15.00	19.00
GSES	31.87	2.36	32.00	24.00	36.00	30.00	34.00
Neuroticism	74.61	21.70	75.00	23.00	128.00	56.00	91.00
Extraversion	120.67	18.41	117.00	70.00	151.00	110.00	136.00
Openness	115.60	12.34	116.00	92.00	141.00	109.00	128.00
Agreeableness	123.25	14.22	127.00	86.00	146.00	118.00	132.00
Conscientiousness	130.93	20.54	133.00	83.00	168.00	117.00	144.00

Legend: M—arithmetic mean; SD—standard deviation; Q25—lower quartile; Q75—upper quartile.

**Table 8 nutrients-16-03294-t008:** Sport discipline and position of health control (MHLC) of the female athletes.

Discipline	*N*	MHLC I Median	Code	Multiple Comparisons with z Tests		
				1	2	3
Football	36	27.00	0	*p* = 0.565	*p* = 1.000	*p* = 0.071
Basketball	32	29.00	1		*p* = 1.000	*p* = 0.000
Handball	68	28.00	2			*p* = 0.002
Volleyball	45	24.00	3			-

**Table 9 nutrients-16-03294-t009:** Discipline and position of health control (MHLC) of the female athletes.

Discipline	*N*	MHLC C Median	Code	Multiple Comparisons with z Tests		
				1	2	3
Football	36	17.50	0	*p* = 1.000	*p* = 0.117	*p* = 1.000
Basketball	32	17.00	1		*p* = 0.064	*p* = 1.000
Handball	68	16.00	2			*p* = 0.416
Volleyball	45	18.00	3			

**Table 10 nutrients-16-03294-t010:** Discipline and generalised self-efficacy (GSES) of the female athletes.

Discipline	*N*	GSES Median	Code	Multiple Comparisons with z Tests		
				1	2	3
Football	36	31.00	0	*p* = 1.000	*p* = 0.018	*p* = 1.000
Basketball	32	31.00	1		*p* = 0.003	*p* = 0.894
Handball	68	33.00	2			*p* = 0.209
Volleyball	45	32.00	3			-

**Table 11 nutrients-16-03294-t011:** Pro-Health Diet Index (nHDI-14) depending on performed sport discipline.

Discipline	*N*	nHDI-14 Median	Code	Multiple Comparisons with z Tests		
				1	2	3
Football	36	13.57	0	*p* = 0.031	*p* = 0.400	*p* = 0.472
Basketball	32	16.04	1		*p* = 0.965	*p* = 1.000
Handball	68	15.21	2			*p* = 1.000
Volleyball	45	14.79	3			-

**Table 12 nutrients-16-03294-t012:** Diet quality indices and position of health control (MHLC), generalised self-efficacy (GSES), personality traits, age and competitive experience of the female athletes (Spearman’s R correlation coefficient).

Variables	*N*	Spearman’s R	t(*N*-2)	*p*-Value
MHLC I and pHDI-10	181	0.13	1.72	0.087
MHLC I and nHDI-14	181	0.08	1.10	0.274
MHLC P and pHDI-10	181	0.08	1.02	0.307
MHLC P and nHDI-14	181	−0.03	−0.36	0.720
MHLC C and pHDI-10	181	−0.02	−0.32	0.750
MHLC C and nHDI-14	181	−0.04	−0.54	0.588
GSES and pHDI-10	181	−0.09	−1.24	0.216
GSES and nHDI-14	181	−0.04	−0.59	0.559
Age and pHDI-10	181	0.14	1.88	0.061
Age and nHDI-14	181	−0.05	−0.65	0.515
Competitive experience and pHDI-10	181	0.14	1.88	0.061
Competitive experience and nHDI-14	181	−0.05	−0.65	0.515
Neuroticism and pHDI-10	181	0.06	0.74	0.461
Extraversion and pHDI-10	181	0.03	0.39	0.695
Openness and pHDI-10	181	−0.22	−3.03	0.003
Agreeableness and pHDI-10	181	−0.00	−0.04	0.966
Conscientiousness and pHDI-10	181	0.13	1.70	0.091
Neuroticism and nHDI-14	181	0.05	0.70	0.485
Extraversion and nHDI-14	181	−0.03	−0.44	0.662
Openness and nHDI-14	181	−0.11	−1.43	0.155
Agreeableness and nHDI-14	181	−0.03	−0.42	0.671
Conscientiousness and nHDI-14	181	−0.10	−1.30	0.195

**Table 13 nutrients-16-03294-t013:** Coefficients of final pro-Health Diet Index (pHDI-10) model.

	Beta	Std. Err. of β	B	Std. Err. of b	t(179)	*p*-Value
Intercept			24.30	3.36	7.23	<0.001
Openness	−0.33	0.08	−0.12	0.03	−4.26	<0.001
Extraversion	0.20	0.08	0.05	0.02	2.64	0.009
Competitive experience	0.18	0.07	0.22	0.09	2.51	0.013

**Table 14 nutrients-16-03294-t014:** Coefficients of final non-Healthy Diet Index (nHDI-14) model.

	Beta	Std. Err. of β	B	Std. Err. of b	t(179)	*p*-Value	*N*
Intercept			15.21	0.429	35.48	<0.001	
Basketball	0.15	0.07	2.03	1.020	1.99	0.048	181

## Data Availability

Data available on request from the authors.

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
