# Peer review of "Selected Determinants of Diet Health Quality among Female Athletes Practising Team Sports"

_nutrients, 2024, doi:10.3390/nu16193294_

Round 1

Reviewer 1 Report

Comments and Suggestions for Authors

The study aim was to conduct an analysis of selected factors  determining diet health quality among a group of elite, Polish sportswomen training team sports. The relationships between age, sports competitive experience, personal resources and personality traits included in the Big Five model and indices of diet health quality (pHDI-10 and nHDI-14) were evaluated. The following research questions were posed:

1) What are the health quality indices of the female athletes’ diet?;

2) What are the personal resources (sense of health control and generalised self-efficacy) and personality traits (neuroticism, extraversion, openness to experience, agreeableness and conscientiousness) of female athletes?;

3) What are the relationships between age, sports experience, personal resources and personality traits as well as the quality of the female athletes’ diet?

The research hypothesis was adopted suggesting that the analysed variables (demographic, sports and psychological) show a relationship with the health quality of the female athletes’ diet, with higher quality of the pro-health diet being supported by: higher age and longer competitive experience, as well as more intense internal health control, higher sense of self-efficacy and extraversion level (related to cheerfulness), conscientiousness (related to dutifulness) and a lower neuroticism level (related to less immoderation).

The manuscript is well structured and deals with a topic that is current and of potential interest to the scientific community. However, I have some suggestions for the authors.

As for the introductory paragraph, it is too long and often the focus of the manuscript is lost. I suggest the authors to make it more concise and to take into consideration some recent studies in which questionnaires were used to evaluate the diet. In this regard, I suggest the authors the following recent publication:

Moscatelli et al., Assessment of Lifestyle, Eating Habits and the Effect of Nutritional Education among Undergraduate Students in Southern Italy, Nutrients. 2023 Jun 26;15(13):2894. doi: 10.3390/nu15132894.

In the tables you must report the units of measurement, for example: cm, year.

The presentation of the results is very detailed; however the presence of many tables could make it difficult to understand the main results. Based on what has been said, if possible, the authors could insert some graphs of the main results.

The beginning of the discussion section should be re-modulated. It would be appropriate to start by highlighting the main results of the study and then discuss them in light of what is present in the literature.

Reviewer 2 Report

Comments and Suggestions for Authors

The authors have submitted the entitled manuscript "Selected Determinants of Diet Health Quality Among Female Athletes Practicing Team Sports" with the aim to analyze selected determinants of diet health quality in a group of elite, Polish female team sports players. Overall, this is well done and the manuscript is very interesting. This is necessary work to have been completed and the authors are commended on this effort.

General Comments For Consideration:

-The conclusion paragraph reads as a list. Please correct this to be a paragraph. Not with the "1." "2." list.

-The authors should consider using Grammarly to ensure any gramatical errors are caught.

-The introduction is quite long (2.5 pages). Can this be shortened? Be concise here if possible.

-Some areas (i.e., methods) need to be adjusted to be in past tense. Please take care to do this throughout the manuscript. For instance, the statement on sample size and using G*Power.

Methods:

A word or two appear to be missing here. "The adopted level of statistical significance α=0.05."

Results:

-Not so sure the wording here works: "The highest percentage of surveyed female athletes consumed fruits....." Does this mean that the respective % for the responses that were highest (i.e., fruits)

-Can the authors provide a bit more clarification of the daily frequency means included in table 5. It is a bit hard to understand (at least initially). Are these means representative of a specific amount of the product? I.e., 0.39 for milk.... just milk consumption in general? Or a specific amount of milk? or does this 0.39 refer to an amount? I ask as in the statistical methods section the authors state that data were represented with minimum, maximum, mean, standard deviation, median, quartiles.

-Please add abbreviations in tables to the table and figure legends so that can be readily recalled by the reader rather than having to scroll up and down to recall these (i.e., DQIP - 3, pHDL-10, etc...).

-Some of the tables have bold lines while others don't. is this intentional?

Reviewer 3 Report

Comments and Suggestions for Authors

This great article examines the actual diet quality in professional sportswomen (team sports only). The introduction is elaborate, as well as the tables do, along with very sound statistics. I have some remarks:

Results: why did you choose the NEO-PI-R over the MMPI as the personality test for this study?

Discussion: could the difference between the team sports be due to the fact that certain sports are typically linked to certain incomes (and thus dietary habits)?
